# Soil Carbon Pool and Carbon Fluxes Estimation in 26 Years after Selective Logging Tropical Forest at Sabah, Malaysia

**Nurul Syakilah Suhaili** [1], **Syazwani Nisa Anuar** [1], **Wilson Vun Chiong Wong** [1], **Daniel Lussetti** [2], **Erik Petter Axelsson** [3], **Niles Hasselquist** [2], **Ulrik Ilstedt** [2] and **Normah Awang Besar** [1,*]

1 Faculty of Tropical Forestry, Universiti Malaysia Sabah, Jalan UMS, Kota Kinabalu 88400, Sabah, Malaysia
2 Department of Forest Ecology and Management, Swedish University of Agricultural Sciences, SE 901 83 Umeå, Sweden
3 Department of Wildlife, Fish, and Environmental Studies, Swedish University of Agricultural Sciences, SE 901 83 Umeå, Sweden
* Correspondence: normabr@ums.edu.my

**Abstract:** The soil carbon pool holds an enormous amount of carbon, making it the largest reservoir in the terrestrial ecosystem. However, there is growing concern that unsustainable logging methods damage the soil ecosystem, thus triggering the release of soil carbon into the atmosphere hence contributing to ongoing climate change. This study uses a replicated ($n = 4$) logging experiment to examine the impact of supervised logging with climber cutting (SLCC) and conventional logging (CL) on basic soil characteristics, litter input to soils, soil carbon pools, and soil respiration in a mixed dipterocarp forest 26 years after logging. This study found that there was no significant difference observed in the soil physicochemical properties and total carbon pools between the logging treatments and the virgin forest. Soil carbon pools dominated the total carbon pools, and the highest mean value was recorded in SLCC ($87.95 \pm 13.67$ Mg C ha$^{-1}$). Conventional logging had a lower mean value ($71.17 \pm 12.09$ Mg C ha$^{-1}$) than virgin forest ($83.20 \pm 11.97$ Mg C ha$^{-1}$). SLCC also shows a higher value of soil respiration rate ($161.75 \pm 21.67$ mg C m$^{-2}$ h$^{-1}$) than CL ($140.54 \pm 12.54$ mg C m$^{-2}$ h$^{-1}$). These findings highlight the importance of accurate quantification of the effect of different logging methods on the forest's carbon pools.

**Keywords:** tropical forest; virgin forest; selective logging; soil carbon pool; litterfall; soil respiration; carbon pool



## 1. Introduction

The soil carbon pool is one of the dominant pools in the terrestrial ecosystem as it can store about 1200 to 1800 Gt (1 Gt = 1 Gigaton = $10^9$ ton) of carbon, which is 3.3 times more than the size of atmospheric carbon (760 Gt) pools and 4.5 times more significant than the biotic carbon pools (560 Gt) [1,2]. Approximately 70% of the global soil organic carbon is stored in the forest ecosystem, and the world soil can potentially sequester around 0.4 to 0.8 Gt carbon per year [3,4]. This vast amount of carbon stored inside the soil carbon pool makes it susceptible to changes and subsequently becomes the carbon source to the atmosphere [5]. Other than negatively affecting soil health and food production, the release of this organic carbon back into the atmosphere could worsen climate change problems that the world currently faces [6].

Logging activity in the forest can also affect the soil respiration rates due to the disturbance in the litter amount and changes in environmental factors such as soil moisture and temperature [7,8]. Soil respiration is one of the largest carbon fluxes in the atmosphere and the primary indicator of soil quality and health [9]. It is also a crucial mechanism in controlling carbon loss from the terrestrial ecosystem [10]. About 98 Gt of carbon is released into the atmosphere through soil respiration, at which a rate that is ten times larger than the emission from the combustion of fossil fuels [11]. Enhancing the knowledge of

the impacts of anthropogenic activity on the spatial variability of soil respiration rates is crucial to carbon balance research and to curtail the rise of carbon dioxide value in the atmosphere [12,13].

The standard logging method that has long been implemented in tropical forests is selective logging [14]. It is a method where only a commercially valuable timber species that meets a particular diameter value will be harvested from the forest. The purpose of leaving the other trees is to maintain the forest cover and to increase the chance of natural tree regeneration in the logged area [15]. This logging method also was introduced to change the clear-cutting method and as a better alternative for reducing the negative impact of logging on the environment [16]. Furthermore, selective logging is considered a sustainable forest management (SFM) practice as it allows people to use the forest resource while maintaining and preserving the forest condition [15].

In the early implementation of selective logging in the forest, there were still some damages that can be observed as it was done in an unsupervised manner, well known as conventional logging [16]. This logging method was conducted without proper planning and guidance to the feller and consequently damaging the residual stand [17]. Reduced impact logging (RIL) practice was then developed to improve the previous logging method. However, it is seldom practiced because of its strict guidelines and greater expense compared to the conventional logging method [18]. Ultimately, a more practical system named supervised logging was introduced. This system was carried out more appropriately than the conventional logging method as it involves directional felling and planned skid trails. The workers also were given detailed instructions before the falling activity was done [16].

The details on the impacts of this logging method on carbon sequestration and pool are still considered scarce in some regions, specifically in Sabah, Malaysia. Previous studies on this area focused more on stand development after logging and the impact of logging on the forest structure [16–19]. Other studies show that a selective logging method could impact soil properties [20], spatiotemporal changes in biomass [21], and the tree composition and diversity in the forest [22]. Here, we report the total soil, organic layer, and litterfall carbon pool in a mixed dipterocarp forest in Sabah, Malaysia, which was experimentally logged using selective logging systems 26 years ago. We also quantified the monthly litterfall production and soil respiration rates across the different logging treatments. The findings are expected to highlight the importance of accurate quantitation of the effect of these anthropogenic activities on the forest's capability to sequester carbon from the atmosphere.

## 2. Materials and Methods

### 2.1. Study Area

This study was conducted in the Swedish University of Agricultural Sciences (SUAS) experimental area, which is located inside Gunung Rara Forest Reserve, Tawau, Sabah, Malaysia (Figure 1). The coordinates of this study area are approximately 4°33′ N, 117°02′ E. This project was carried out in 1992 to investigate the effect of logging on forest recovery. In addition, this project covered about 3000 hectares and is currently managed by the Yayasan Sabah group as one of their Forest Management Areas. Gunung Rara Forest Reserve is a virgin tropical rainforest dominated by dipterocarp trees. About 230 tree species were identified, and this forest was gazetted as Virgin Jungle (Class VI) under the Forest Enactment 1968 [18].

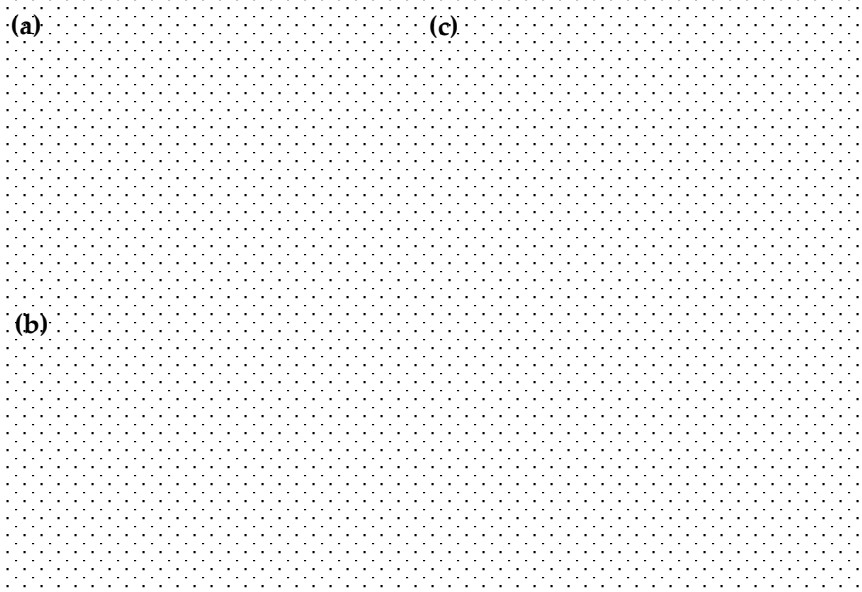

**Figure 1.** (**a**) Map of Malaysia. (**b**) Map of Sabah, Malaysia, showing the location of study area established inside Gunung Rara Forest Reserve. (**c**) Plot layout of the study area. The legend indicates: VF = virgin forest (as the unlogged forest), CL = conventional logging, SLCC = supervised logging with climber cutting.

This area's altitude ranges between 300 and 600 m above sea level and the soil is classified as Orthic Acrisol. The study area has an average annual rainfall ranging from 2700 to 3400 mm per year, and its annual temperature is 27 °C (Figure 2).

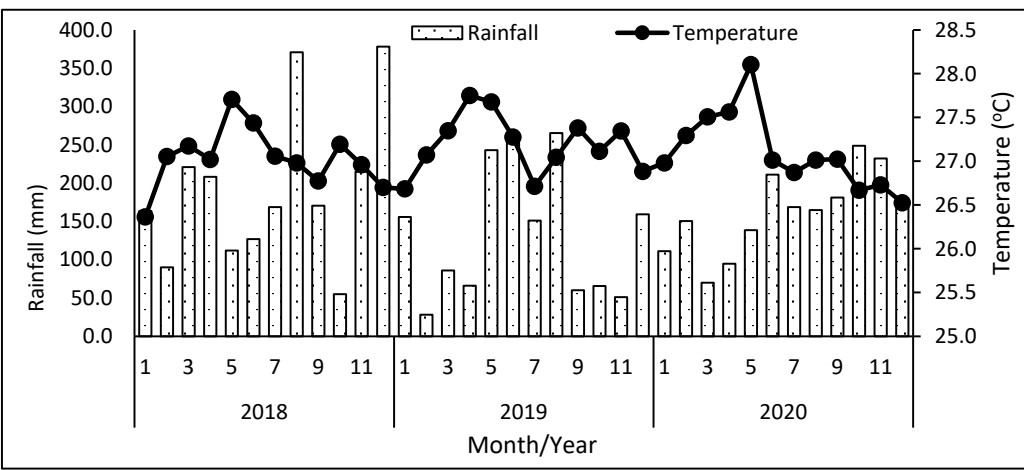

**Figure 2.** Monthly mean rainfall (mm) and temperature (°C) at study site, 2018–2020. The data was provided by the Malaysian Meteorological Department [23].

### 2.2. Experimental Design

The plot establishment was started in March through June 1992. Then, the harvesting started in June 1993 and finished in August of the same year. The summary of stand characteristics in all sampling areas is shown in Table 1. The plot size for this study was 60 × 60 m (0.36 ha). The experiment was designed using a randomized 2 × 2 factorial complete block design. The average inclination of each net plot was taken into consideration for the blocking factor, which varied from 4.1° to 24.7°. Two types of logging treatments studied in this research are supervised logging with climber cutting (SLCC) and conventional logging (CL). In conventional logging, the trees can be fallen before the presence of the crawler tractor. There were no guidelines given to the feller on the felling technique. The workers

were usually the contractor's personnel and did not have any formal training and education on the harvesting operation [17].

**Table 1.** The summary of stand characteristics in supervised logging with climber cutting plots, conventional logging plots, and virgin forest plots, before and after being logged.

| Study Areas | Stand Density (Trees ha$^{-1}$) | | | Mean DBH (cm) | | | Basal Area (m$^2$ ha$^{-1}$) | | | Aboveground Biomass (Mg ha$^{-1}$) * | | |
|---|---|---|---|---|---|---|---|---|---|---|---|---|
| | 1992 | 1993 | 2017 | 1992 | 1993 | 2017 | 1992 | 1993 | 2017 | 1992 | 1993 | 2017 |
| SLCC [1] | 519 | 420 | 532 | 22.57 | 22.01 | 24.31 | 32.67 | 23.14 | 35.54 | 266.20 | 182.51 | 285.66 |
| CL [2] | 509 | 397 | 538 | 23.35 | 23.34 | 23.99 | 33.63 | 26.08 | 37.31 | 278.11 | 213.03 | 307.02 |
| VF [3] | 535 | 534 | 518 | 23.24 | 23.04 | 23.81 | 35.96 | 35.30 | 37.35 | 295.66 | 290.29 | 311.13 |

[1] Supervised logging with climber cutting; [2] conventional logging; [3] virgin forest; * calculated using Basuki et al. (2009). The values represent mean of the measurements.

A more appropriate way was applied in supervised logging treatment, where the trees were extracted after a skid trail was established. The trail was built systematically aligned to each other and tractors were not allowed to open a new trail to skid the logs. The potential crop trees with DBH ranging from 40–59 cm found along the skid trail were marked to avoid any damage to them. The trees were harvested by felling them toward the skid trail direction. This method minimizes the impact on the trees nearby. In contrast, trees that cannot be directed into the skid trail fell in a direction that would cause minor damage to other trees [16].

The supervised logging treatment was combined with the preharvest climber cutting treatment. It is well known that climber plants such as lianas connect trees and could cause damage to the connected trees when the harvesting activity was carried out. The effect could be reduced by removing it before harvesting the timber [18].

The plot for each treatment was replicated into four plots. A virgin forest (VF) as an unlogged forest was used as the control plot. The total plot for this study is 12 plots. The field data collection for soil and the organic layer was performed in January 2019 and September 2019, while litterfall collection and soil respiration measurements were carried out from March 2019 until February 2020. Soil sampling was performed at four random points, while the organic layer was collected at nine random points inside the plots. The litterfall was collected from six traps and soil respiration was measured from eight soil respiration chambers.

*2.3. Soil Carbon Pool Estimation*

Two types of soil samples were taken: undisturbed soil for bulk density and mixed soil samples for soil properties analysis. The soil was collected at four random points within each plot. All samples were taken at four different depths: 0–10, 10–20, 20–50, and 50–100 cm, and four soil samples were taken on each layer for four plot replicates. The total of soil samples was 16 for each layer. The soil bulk density samples were collected using a cylinder with a volume of 98.125 cm$^3$. The samples were put into a plastic sample container and labeled by their plot and layer, then brought into the laboratory for further analysis.

The soil physicochemical properties measured in this study were bulk density, texture, pH, percentage of organic matter, and carbon and nitrogen concentration. Soil bulk density was expressed as the dry mass ratio over its volume [24]. Soil texture was determined using the pipette method described in [25] and the USDA Soil Classification Triangle. The soil water suspension (1:2.5) method was used to measure soil pH [26]. The soil pH was determined using a pH meter (Ohaus Corporation, Parsippany, NJ, USA) after the solution was shaken using a shaker machine for 30 minutes. This pH meter was calibrated using pH 4.0, 7.0, and 9.0 buffer solutions before the analysis. The percentage of soil organic matter was determined using the loss-on-ignition method described in [27], where the samples were ignited at 500 °C for 24 hours in a furnace. A Vario Max CN Elemental Analyser

(Elementar Analysensysteme, Langenselbold, Germany) analyzed the concentration of carbon and nitrogen.

The soil carbon pool was calculated by using the information from soil bulk density (BD), soil depth interval (SDI), and soil carbon concentration (C). The calculation used the equation given below:

$$C_{Soil} \text{ (Mg C ha}^{-1}) = BD \text{ (g cm}^{-3}) \times SDI \text{ (cm)} \times C \text{ (\%)} \tag{1}$$

The value for each layer was then summed to obtain the total soil carbon pool per hectare [28].

### 2.4. Organic-Layer Carbon-Pool Estimation

The samples for the organic layer were collected at nine random points within the plots using a $0.5 \times 0.5$ m ($0.25$ m$^2$) sampling frame. All forest litter within the sampling frame, such as small branches, leaves, and fruits, was collected and stored inside a plastic sample container. All samples were oven-dried at 70 °C for 72 hours (or until constant weight) in the laboratory to estimate their biomass. The carbon was then estimated using a 0.5 conversion factor, as carbon in plants is 50% of their biomass [28].

### 2.5. Litterfall Carbon Pool Estimation

The litterfall traps were built using four small blocks of wood as the stands and a nylon mesh as the traps. Six litterfall traps were installed inside all plots. The trap size was $0.5 \times 0.5$ m ($0.25$ m$^2$). Sampling was performed from March 2019 until February 2020 and collected once every month. All forest litter was collected, such as leaves, fruits, and small branches that fell into the traps. All samples were oven-dried at 70 °C for 72 hours (or until constant weight) in the laboratory to estimate their biomass. The carbon was then estimated using a 0.5 conversion factor, as the carbon for plants is 50% of their biomass [28].

### 2.6. Soil Respiration Measurement

Soil respiration was measured using the Vaisala CARBOCARP® Carbon Dioxide Probe GMP343 and the Vaisala Handheld Measurement Indicator MI70 (Vaisala, Finland). In addition, a PVC collar with an average volume of 0.00567 m$^2$ (diameter 0.23 m and height 0.14 m) was installed approximately 4 cm inside the soil to prevent air loss during measurement.

Furthermore, eight soil respiration chambers were installed within each plot. The collar also was installed on the undisturbed forest floor area and remained on-site throughout the measurement period. This measurement was performed the same as for the litterfall, and was recorded once a month for twelve months. The data collection started on March 2019 and finished in February 2020.

### 2.7. Statistical Analysis

One-way analysis of variance (ANOVA) with a post hoc test using Tukey's test at a significant value of less than 0.05 was performed to investigate the statistical differences between the logging treatments and virgin forest. The statistical analysis was carried out using IBM SPSS Statistic 24 statistical software.

## 3. Results

### 3.1. Soil Physicochemical Properties

Tables 2 and 3 show the soil's physicochemical properties in the study areas. The soil bulk density, which is vital information for the soil carbon pool, ranges between $1.07 \pm 0.08$ g cm$^{-3}$ and $1.46 \pm 0.05$ g cm$^{-3}$. The highest mean value was recorded in the conventional logging areas at the deepest sampling depth (50–100 cm), while the lowest mean value was recorded on the surface (0–10 cm) in virgin forest areas. The trend of soil bulk density shows it decreases with depth. No significant difference ($p = 0.266$) was found between the mean of soil bulk density across the different sampling areas. There were four

types of soil texture recorded in this study: clay, sandy loam, sandy clay loam, and sandy clay. Based on Table 2, clay and sand dominated the soil particles, while silt was the least recorded particle. Sand represents 30% to 72% of the proportion, while clay represents 3% to 47%. On the other hand, silt only represents 3% to 26% of the proportion.

Table 3 shows the soil's chemical properties and carbon and nitrogen concentrations across the different logging treatments and virgin forests. All soil in the study areas was an acidic type of soil. The pH (Table 3) ranged from $3.87 \pm 0.09$ to $4.54 \pm 0.15$. The most acidic soil was found on the surface layer (0–10 cm) of the virgin forest, while the least acidic soil was found in the fourth layer (50–100 cm) in conventional logging areas. The statistical analysis of one-way ANOVA showed that no significant differences ($p = 0.159$) were found between the mean soil pH across the different study areas.

The soil organic matter (Table 3) ranged from $4.15 \pm 0.55\%$ to $7.40 \pm 1.06\%$, where the lowest percentage was recorded for the fourth depth (50–100 cm) in the conventional logging areas while the highest percentage was recorded for the surface (0–10 cm) of the supervised logging with climber cutting areas. It is also recorded that the percentage of soil organic matter decreased as the depth increased. No significant differences ($p = 0.262$) were found between the mean of soil organic matter across the different study areas.

Both carbon and nitrogen concentrations (Table 3) showed a similar trend across the sampling depth, which decreased as the depth increased. The lowest value of carbon concentration was recorded for the 50–100 cm soil depth of the conventional logging plots, with a mean value of $0.30 \pm 0.08\%$, while the highest value was recorded for the 0–10 cm soil depth in control areas, with a mean value of $1.99 \pm 0.31\%$. For nitrogen, the mean ranged from $0.02 \pm 0.01\%$ to $0.18 \pm 0.05\%$, in which the lowest was recorded for the fourth layer (50–100 cm) in the conventional logging and virgin forest areas while the highest was recorded for the first layer (0–10 cm) in the supervised logging with climber cutting areas. However, no significant differences ($p = 0.455$) were found between the mean concentrations of carbon and nitrogen across the different sampling areas.

### 3.2. Litterfall Production

The monthly variation in litterfall biomass in the supervised logging with climber cutting, conventional logging, and virgin forest areas is shown in Figure 3.

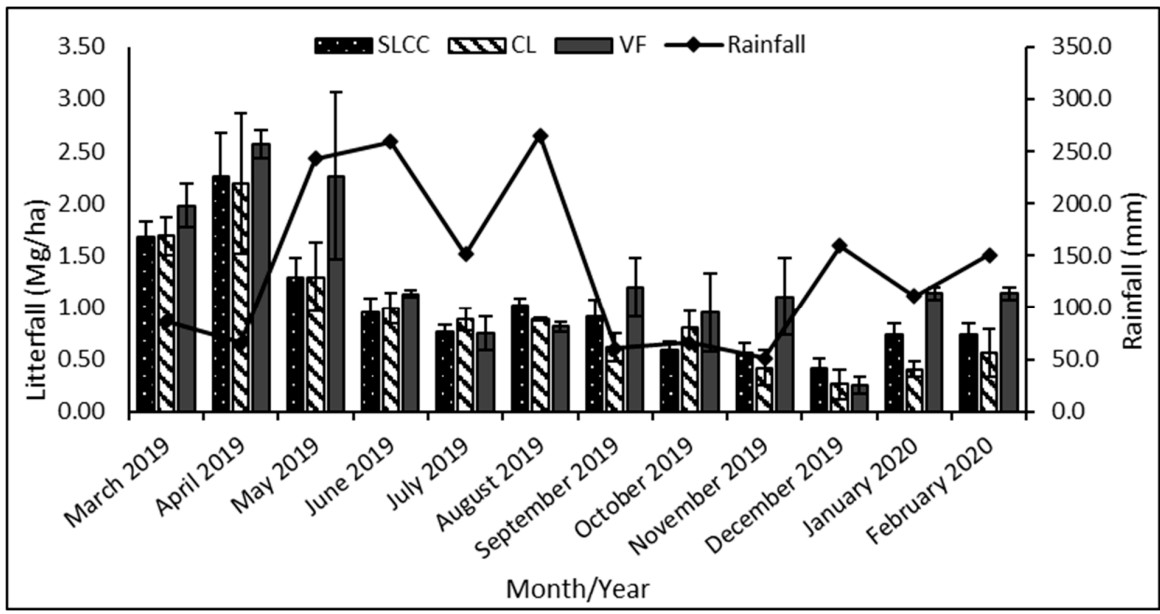

**Figure 3.** Mean monthly litterfall (Mg ha$^{-1}$) production in supervised logging with climber cutting plots (SLCC), conventional logging plots (CL), and virgin forest plots (VF) at Gunung Rara Forest Reserve, Sabah, Malaysia, from March 2019–February 2020. Error bars represent the standard error for the measurements.

**Table 2.** Soil physical properties at four depths (0–100 cm) in supervised logging with climber cutting plots (SLCC), conventional logging plots (CL), and virgin forest plots (VF).

| Depth (cm) | Bulk Density (g cm$^{-3}$) | | | Clay (%) | | | Silt (%) | | | Sand (%) | | | Soil Texture | | |
|---|---|---|---|---|---|---|---|---|---|---|---|---|---|---|---|
| | SLCC [1] | CL [2] | VF [3] | SLCC [1] | CL [2] | VF [3] | SLCC [1] | CL [2] | VF [3] | SLCC [1] | CL [2] | VF [3] | SLCC [1] | CL [2] | VF [3] |
| 0–10 | 1.19 ± 0.03 a | 1.22 ± 0.05 a | 1.07 ± 0.08 a | 52 ± 5.32 | 26 ± 1.28 | 29 ± 4.13 | 8 ± 1.22 | 3 ± 0.98 | 13 ± 1.97 | 33 ± 1.11 | 71 ± 2.76 | 54 ± 5.67 | Clay | Sandy Clay Loam | Sandy Clay Loam |
| 10–20 | 1.29 ± 0.03 a | 1.30 ± 0.04 a | 1.22 ± 0.11 a | 3 ± 1.22 | 31 ± 4.89 | 29 ± 3.18 | 21 ± 2.31 | 3 ± 0.98 | 5 ± 2.11 | 69 ± 4.76 | 65 ± 2.75 | 64 ± 3.48 | Sandy Loam | Sandy Clay Loam | Sandy Clay Loam |
| 20–50 | 1.34 ± 0.03 a | 1.43 ± 0.03 a | 1.29 ± 0.05 a | 23 ± 1.43 | 21 ± 4.12 | 44 ± 2.19 | 10 ± 1.32 | 8 ± 1.01 | 8 ± 2.93 | 65 ± 3.85 | 69 ± 2.91 | 44 ± 3.29 | Sandy Clay Loam | Sandy Clay Loam | Clay |
| 50–100 | 1.44 ± 0.02 a | 1.46 ± 0.05 a | 1.42 ± 0.02 a | 8 ± 0.79 | 26 ± 4.10 | 47 ± 2.87 | 26 ± 1.54 | 5 ± 1.63 | 3 ± 1.76 | 65 ± 3.29 | 67 ± 1.65 | 43 ± 3.88 | Sandy Loam | Sandy Clay Loam | Clay |

[1] Supervised logging with climber cutting; [2] conventional logging; [3] virgin forest. The values stand for mean ± standard error of the measurements. The same letter within the same row shows there were no significant differences ($p > 0.05$) between the mean of measurements across the different study plots, based on Tukey's test.

**Table 3.** Soil chemical properties and carbon and nitrogen concentration at four depths (0–100 cm) in supervised logging with climber cutting plots (SLCC), conventional logging plots (CL), and virgin forest plots (VF).

| Depth (cm) | pH Value | | | Organic Matter (%) | | | Carbon (%) | | | Nitrogen (%) | | | C:N | | |
|---|---|---|---|---|---|---|---|---|---|---|---|---|---|---|---|
| | SLCC [1] | CL [2] | VF [3] | SLCC [1] | CL [2] | VF [3] | SLCC [1] | CL [2] | VF [3] | SLCC [1] | CL [2] | VF [3] | SLCC [1] | CL [2] | VF [3] |
| 0–10 | 4.13 ± 0.26 a | 4.36 ± 0.12 a | 3.87 ± 0.09 a | 7.40 ± 1.06 a | 4.98 ± 0.40 a | 7.33 ± 0.59 a | 1.73 ± 0.24 a | 1.31 ± 0.16 a | 1.99 ± 0.31 a | 0.18 ± 0.05 a | 0.11 ± 0.04 a | 0.17 ± 0.04 a | 13.07 ± 4.37 a | 15.22 ± 2.71 a | 12.91 ± 2.24 a |
| 10–20 | 4.38 ± 0.21 a | 4.50 ± 0.07 a | 4.00 ± 0.16 a | 6.00 ± 1.27 a | 4.59 ± 0.54 a | 5.89 ± 0.70 a | 0.95 ± 0.18 a | 0.91 ± 0.23 a | 1.01 ± 0.17 a | 0.07 ± 0.02 a | 0.08 ± 0.04 a | 0.05 ± 0.01 a | 15.59 ± 3.11 a | 16.25 ± 2.82 a | 20.98 ± 2.06 a |
| 20–50 | 4.42 ± 0.16 a | 4.41 ± 0.18 a | 4.24 ± 0.08 a | 6.07 ± 1.08 a | 4.22 ± 0.59 a | 5.27 ± 0.48 a | 0.64 ± 0.11 a | 0.52 ± 0.09 a | 0.68 ± 0.16 a | 0.04 ± 0.13 a | 0.03 ± 0.01 a | 0.04 ± 0.01 a | 18.38 ± 5.66 a | 19.74 ± 2.57 a | 18.94 ± 2.73 a |
| 50–100 | 4.21 ± 0.09 a | 4.54 ± 0.15 a | 4.39 ± 0.02 a | 5.93 ± 1.08 a | 4.15 ± 0.55 a | 5.06 ± 0.68 a | 0.42 ± 0.09 a | 0.30 ± 0.08 a | 0.36 ± 0.05 a | 0.04 ± 0.01 a | 0.02 ± 0.01 a | 0.02 ± 0.01 a | 12.61 ± 2.89 a | 16.05 ± 1.28 a | 17.31 ± 1.26 a |

[1] Supervised logging with climber cutting; [2] conventional logging; [3] virgin forest. The values stand for mean ± standard error of the measurements. The same letter within the same row shows there were no significant differences ($p > 0.05$) between the mean of measurements across the different study plots, based on Tukey's test.

The virgin forest plots recorded the highest and lowest monthly litterfall production, $0.25 \pm 0.08$ and $2.56 \pm 0.14$ Mg ha$^{-1}$. On the other hand, the monthly litterfall biomass production range for the other study plots was between $0.41 \pm 0.09$ and $2.25 \pm 0.42$ Mg ha$^{-1}$ for supervised logging with climber cutting plots and $0.26 \pm 0.14$ and $2.19 \pm 0.67$ Mg ha$^{-1}$ for conventional logging plots. The peak litterfall production for all plots was recorded in April 2019 and the lowest was recorded in December 2019.

Though all plots showed a similar pattern in litterfall biomass production throughout the year, one-way ANOVA statistical analysis showed a significant difference between the mean litterfall production in August 2019 and January 2020. The finding shows that in August 2019, the supervised logging with climber cutting areas produced significantly higher litterfall biomass than the virgin forest area. The values were $1.03 \pm 0.07$ and $0.82 \pm 0.05$ Mg ha$^{-1}$, respectively.

On the contrary, the conventional logging plots produced a similar amount of litterfall biomass in both study areas on that month, which was $0.89 \pm 0.01$ Mg ha$^{-1}$. Next, the litterfall production in January 2020 showed the opposite result from August 2019, in which the virgin forest areas produced significantly higher litterfall biomass than supervised logging with climber cutting areas and the conventional areas. The values were $1.14 \pm 0.06$ Mg ha$^{-1}$ for control areas, $0.73 \pm 0.11$ Mg ha$^{-1}$ for supervised logging with climber cutting areas, and lastly $0.40 \pm 0.08$ Mg ha$^{-1}$ for conventional logging areas.

### 3.3. Soil Carbon Pool

Table 4 shows the total mean of soil carbon pool in supervised logging with climber cutting areas, conventional logging areas, and virgin forest areas. Supervised logging with climber cutting areas recorded the highest value of soil carbon pool with a mean total of $87.95 \pm 13.67$ Mg C ha$^{-1}$. This value was followed by the virgin forest areas with $83.20 \pm 11.96$ Mg C ha$^{-1}$. Conversely, the lowest amount of soil carbon pool was recorded in conventional logging with $71.17 \pm 12.09$ Mg C ha$^{-1}$. The one-way ANOVA analysis shows no significant differences ($p = 0.639$) in the total mean of soil carbon pool across the different study areas.

**Table 4.** The total mean of soil carbon pool (Mg C ha$^{-1}$) at four different depths (0–100 cm) in supervised logging with climber cutting areas (SLCC), conventional logging areas (CL), and virgin forest areas (VF).

| Study Areas | Soil Carbon Pool (Mg C ha$^{-1}$) | | | | Total Soil Carbon Pool (Mg C ha$^{-1}$) |
|---|---|---|---|---|---|
| | 0–10 cm | 10–20 cm | 20–50 cm | 50–100 cm | |
| SLCC [1] | $20.49 \pm 2.56$ a | $12.13 \pm 2.09$ a | $25.22 \pm 3.99$ a | $30.12 \pm 5.87$ a | $87.95 \pm 13.67$ a |
| CL [2] | $15.84 \pm 1.62$ a | $11.74 \pm 2.70$ a | $21.96 \pm 3.48$ a | $21.63 \pm 5.50$ a | $71.17 \pm 12.09$ a |
| VF [3] | $20.76 \pm 2.50$ a | $11.97 \pm 1.57$ a | $25.46 \pm 5.30$ a | $25.02 \pm 3.57$ a | $83.20 \pm 11.97$ a |

[1] Supervised logging with climber cutting; [2] conventional logging; [3] virgin forest. The values represent the mean ± standard error of the measurements. The same letter within the same column shows there were no significant differences ($p > 0.05$) between the mean of measurements across the different study plots, based on Tukey's test.

Table 4 also shows the trend of soil carbon pool by its depth. The lowest range of soil carbon pool was recorded for the 10–20 cm depth, between $11.73 \pm 2.70$ and $12.13 \pm 2.09$ Mg C ha$^{-1}$. The range of soil carbon pool in the top 10 cm was between $15.84 \pm 1.62$ and $20.76 \pm 2.50$ Mg C ha$^{-1}$. The highest range was recorded for the deepest depth of the sampling area, which is the 50–100 cm soil depth, with values between $21.63 \pm 5.49$ and $30.12 \pm 5.87$ Mg C ha$^{-1}$. No statistical differences ($p > 0.05$) were found in between the soil carbon pool at the same depth across the different study areas.

### 3.4. Total Carbon Pool (Organic Layer, Litterfall, and Soil Carbon Pool)

Table 5 shows the total carbon pools comprising the organic layer, litterfall, and soil. Soil carbon pool had the highest range of total carbon pools, from $71.17 \pm 12.09$ to

$87.95 \pm 13.67$ Mg C ha$^{-1}$, followed by litterfall carbon from $5.50 \pm 0.39$ to $7.64$ Mg C ha$^{-1}$. The organic layer carbon pool contributed the least with ranges from $1.28 \pm 0.17$ to $1.58 \pm 0.34$ Mg C ha$^{-1}$. No significant difference was observed between the total carbon pools of the organic layer and soil across the study area. In contrast, the statistical analysis one-way ANOVA shows that the litterfall carbon pool had statistically ($p = 0.008$) produced a higher amount of total carbon pools than the other areas.

**Table 5.** Total carbon pools (organic layer, litterfall, and soil) in supervised logging with climber cutting areas (SLCC), conventional logging areas (CL), and virgin forest areas (VF).

| Study Areas | Carbon Pools (Mg C ha$^{-1}$) | | | Total Carbon Pools (Mg C ha$^{-1}$) |
|---|---|---|---|---|
| | **Organic Layer** | **Litterfall** | **Soil** | |
| SLCC [1] | $1.28 \pm 0.17$ a | $5.95 \pm 0.20$ a | $87.95 \pm 13.67$ a | $95.17 \pm 13.66$ a |
| CL [2] | $2.00 \pm 0.27$ a | $5.50 \pm 0.39$ a | $71.17 \pm 12.09$ a | $78.66 \pm 11.92$ a |
| VF [3] | $1.58 \pm 0.34$ a | $7.64 \pm 0.36$ b | $83.20 \pm 11.97$ a | $92.41 \pm 13.69$ a |

[1] Supervised logging with climber cutting; [2] conventional logging; [3] virgin forest. The values represent the mean $\pm$ standard error of the measurements. The same letter within the same column shows there were no significant differences ($p > 0.05$) between the mean of measurements across the different study plots, based on Tukey's test.

This study's supervised logging with climber cutting plots stored the highest total carbon pool (organic layer, litterfall, and soil) with $95.17 \pm 13.66$ Mg C ha$^{-1}$. In addition, it stores more carbon than the virgin forest plots with $92.41 \pm 13.69$ Mg C ha$^{-1}$. On the other hand, conventional logging had the lowest amount of total carbon pools with $78.66 \pm 11.92$ Mg C ha$^{-1}$. However, no significant difference ($p = 0.661$) was observed in the total carbon pools across the study areas.

*3.5. Soil Respiration Rates*

Figure 4 shows the monthly variation in soil respiration in supervised logging with climber cutting, conventional logging, and virgin forest plots.

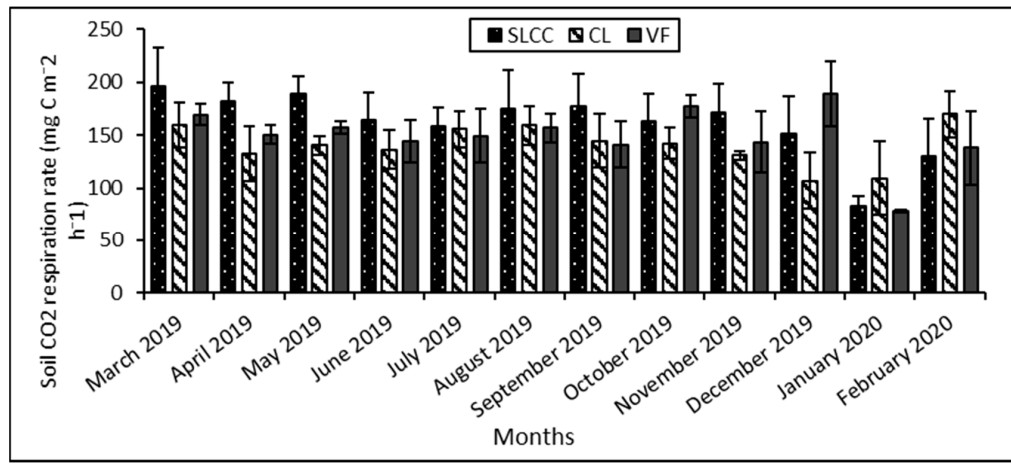

**Figure 4.** Mean monthly soil respiration (mg C m$^{-2}$ h$^{-1}$) measurements in supervised logging with climber cutting areas (SLCC), conventional logging areas (CL), and virgin forest areas (VF) at Gunung Rara Forest Reserve, Sabah, Malaysia, from March 2019 to February 2020. Error bars represent the standard error for the measurements.

Seasonal soil respiration varied throughout the year, with a mean range of $78.35 \pm 1.07$ to $196.47 \pm 36.46$ mg C m$^{-2}$ h$^{-1}$. The lowest mean value was recorded at the virgin forest areas in January 2020, while the highest was at the supervised logging with climber cutting areas in March 2019. The supervised logging with climber cutting areas and the conventional logging plots showed a similar trend of soil respiration throughout the year.

The mean range for the areas was between $83.02 \pm 8.77$ and $196.47 \pm 36.46$ mg C m$^{-2}$ h$^{-1}$ and $106.86 \pm 26.89$ and $169.95 \pm 21.77$ mg C m$^{-2}$ h$^{-1}$, respectively.

Whereas the soil respiration measurements at the virgin forest plots were consistent from March 2019 to September 2019 with a mean value of $169.18 \pm 10.04$ to $141.07 \pm 22.09$ mg C m$^{-2}$ h$^{-1}$, but then fluctuated significantly from October 2019 until February 2020. During these periods, it recorded two peaks: in October 2019, with a mean value of $177.38 \pm 10.21$ mg C m$^{-2}$ h$^{-1}$, and in December 2019, with a mean value of $189.11 \pm 30.43$ mg C m$^{-2}$ h$^{-1}$. The recording then dropped to its lowest point in January 2020 with a mean value of $78.35 \pm 1.07$ mg C m$^{-2}$ h$^{-1}$ and then rose somewhat in the last month of sampling with a mean value of $138.05 \pm 34.51$ mg C m$^{-2}$ h$^{-1}$.

The supervised logging with climber cutting areas showed a higher annual soil respiration rate than the other areas (Figure 5). The mean value was $161.75 \pm 21.67$ mg C m$^{-2}$ h$^{-1}$. The virgin forest areas then followed this with a mean value of $149.59 \pm 12.46$ mg C m$^{-2}$ h$^{-1}$. On the other hand, the conventional logging areas have the lowest annual soil respiration rate with a mean value of $140.54 \pm 12.54$ mg C m$^{-2}$ h$^{-1}$. However, the statistical analysis of one-way ANOVA showed no significant differences ($p = 0.671$) between the mean value of annual soil respiration rate across the different logging treatments and virgin forests.

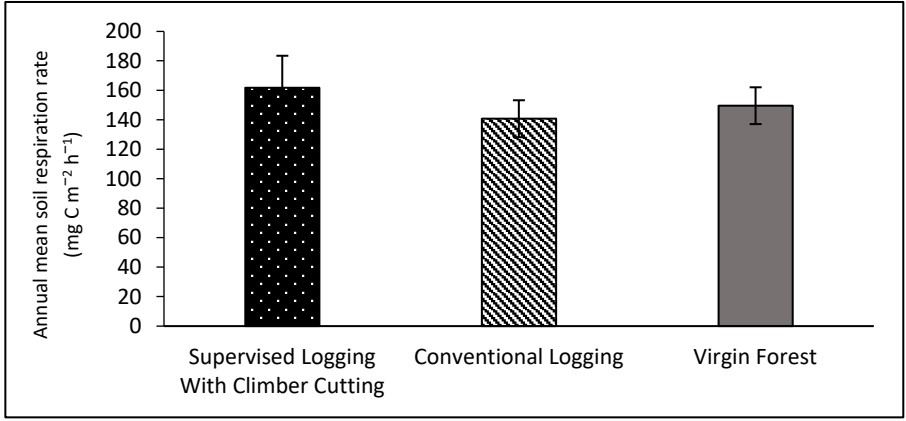

**Figure 5.** The annual soil respiration rate (mg C m$^{-2}$ h$^{-1}$) in supervised logging with climber cutting plots (SLCC), conventional logging plots (CL), and virgin forest plots (VF) at Gunung Rara Forest Reserve, Sabah, Malaysia. Error bars represent the standard error for the measurements.

## 4. Discussion

### 4.1. Soil Physicochemical Properties of the Forest 26 Years after Being Logged

Soil physical properties are one of the crucial indicators of soil quality in a forest ecosystem other than being the source of moisture and nutrients essential for the plant to grow [29,30]. After 26 years of being logged, both treatment plots showed no significant difference in their soil properties when compared with the virgin forest plots. The result for soil bulk density (Table 2) showed that its mean value tends to increase as the logging intensities increase. This trend is due to the variation in the percentage of soil organic matter, soil porosity, and soil compaction [31]. Less soil organic matter was available in the subsurface layers of the soil, subsequently making it more compacted and less aggregated. Hence, this explains the increasing value of soil bulk density as the depth increases [32].

Sand dominated the soil texture in this study (Table 2). The range for sand was from $33 \pm 1.11\%$ to $71 \pm 2.76\%$, while the range for clay and silt was from $3 \pm 1.22$ to $52 \pm 5.32$ and $3 \pm 0.98$ to $26 \pm 1.54$, respectively. The percentage of sand is directly proportional to the mean of soil bulk density, hence explaining the high value of soil bulk density [33]. Soil carbon and nitrogen concentration, nutrient content, permeability, structure, and porosity are other properties correlated with soil texture [34].

The soil in all study plots is acidic with a range from $3.87 \pm 0.09$ to $4.54 \pm 0.15$ (Table 3). This acidity is due to exposure to many sources of organic matter, such as tree litter and dead animals [35,36]. This condition is also somewhat necessary for the forest's soil so the

plants can receive sufficient nutrients from the soil [31]. Soil organic matter is a vital source of plant nutrients and carbon. Thus, adding it could enhance soil carbon sequestration from the atmosphere [37,38]. In addition, it could influence soil water-holding capacity, compaction, aggregation, and cation exchange capacity (CEC) [39,40].

Moreover, the canopy opening caused by the high intensity of logging increases light penetration to the forest floor, raising the temperature [20]. This activity enhances the decomposition of soil organic matter and releases carbon into the atmosphere [40]. The loss of soil organic matter signifies the depletion of carbon pools inside the forest, making it crucial to protect its source so its role as a carbon reservoir can be balanced and maintained [20].

### 4.2. Litterfall and Organic Layer Carbon Pools

Litter decomposition on the surface of the forest floor is an essential element in the forest ecosystem as it is the source of organic matter and nutrients for the soil [41]. The result shows conventional logging treatment has a higher mean of organic layer biomass than supervised logging with climber cutting treatment and the virgin forest plot (Table 5). This contribution is probably because of the young stand in the conventional logging as it tends to shed more litter than the stand in the old forest, such as the intact forest and the virgin jungle [42]. In their study of naturally regenerated *Acacia mangium* stands in Bangkok, Pitas, Hung et al. [43] recorded a similar mean value of fine litter biomass, which was 4.20 Mg ha$^{-1}$.

Compared to this study, results in Suhaili et al. [2], reported for a tropical montane forest in Sabah, Borneo, showed a higher value of organic layer biomass. The values were $6.59 \pm 0.05$ Mg ha$^{-1}$ in the intact forest and $6.50 \pm 0.05$ Mg ha$^{-1}$ in the logged-over forest. The differences between the study sites and different altitudes show that temperature plays an essential factor in the accumulation of organic matter on the surface of the forest floor; higher altitudes usually have lower temperatures than the lowland forest. These differences make the decomposition process slower and could preserve more litter mass and carbon pool [36].

Intact forests, or virgin forests in general, could produce a higher amount of accumulated litterfall compared to the logged forest [2]. The findings show that virgin forest areas have recorded significantly higher annual litterfall biomass than the treatments area (Table 5). Compared to this study, the results in Suhaili et al. [2], reported for a tropical montane forest in Borneo, Sabah, indicate a lower annual biomass production range. The values were $6.59 \pm 0.05$ Mg ha$^{-1}$ yr$^{-1}$ for the intact forest, $6.50 \pm 0.05$ Mg ha$^{-1}$ yr$^{-1}$ for the logged-over forest, and $7.41 \pm 0.07$ Mg ha$^{-1}$ yr$^{-1}$ for the plantation forest. In addition, Dent et al. [44], in their study at Kabili–Sepilok Forest Reserve, Sabah, Malaysia also recorded a lower amount of annual litterfall biomass production compared to this study site, which ranged between $5.70 \pm 0.12$ and $7.70 \pm 0.23$ Mg ha$^{-1}$ yr$^{-1}$.

In a study at a forest plantation in Gum Gum Forest Reserve, Sabah, Malaysia, Inagaki et al. [45] recorded a similar range of annual litterfall biomass production, which was from $6.2 \pm 0.29$ to 13.5 Mg ha$^{-1}$ yr$^{-1}$. Jasinska et al. [46] lists a few factors that could influence litterfall mass, structure, and chemical composition. Some factors are species composition, stage of succession developments, forest-age habitat conditions, and forest regeneration form [42,47]. These factors explain the differences in the mean production of litterfall biomass across the different logging treatments and the different land use types. Rainfall distribution also influenced the production of litterfall biomass, as its absence could cause the trees to shed more leaves due to drought stress [2].

### 4.3. Soil Carbon Pool

The finding (Table 4) shows that after 26 years of logging, there is no significant difference observed in the total soil carbon pools between the treatment areas and the virgin forest (unlogged forest). The mean value of the supervised logging with climber cutting areas ($87.95 \pm 13.67$ Mg C ha$^{-1}$) showed that it had higher soil carbon than conventional

logging areas (71.17 $\pm$ 12.09 Mg C ha$^{-1}$) and virgin forest areas (83.20 $\pm$ 11.97 Mg C ha$^{-1}$). Conventional logging areas still have a lower mean value of total soil carbon pool than the virgin forest areas, despite no significant differences observed. This finding suggests that the recovery process in these areas is much slower than supervised logging with climber cutting treatments. However, a more detailed study must fill the knowledge gap on this discovery. As this research was done 26 years after the logging activities, there is no longer a significant difference observed between the treatment plots and the unlogged ones. More plots need to be established and an early sampling performed in the future to observe the recovery period of the soil.

The finding in this study also is in line with that reported in Ngo et al. [48], where the soil carbon pool was higher in the secondary forest (103.9 Mg C ha$^{-1}$) compared to the primary forest (77.5 Mg C ha$^{-1}$). They also stated that the increasing value of the soil carbon pool in the secondary forest might be influenced by the increasing value of carbon concentrations in the soil. Furthermore, the higher value of carbon: nitrogen ratios indicate there was less decomposition of organic matter in that area.

The value of the soil carbon pool stored in this study area was found to be higher than the value recorded by Besar et al. [49] for the natural forest at Sabah, Malaysia, which was 36.30 $\pm$ 4.74 Mg C ha$^{-1}$. They also found that the agroforestry system, a combination of oil palm and agarwood, can increase the total soil carbon pool (0–30 cm). Their result suggests that proper and suitable forest management can enhance the efficiency of soil carbon sequestration in the forest even after being logged several decades ago [49]. Other factors that can influence the value of the soil organic carbon pool in an area are the sampling depth, soil type, topography, climate, and type of land use [2,49]. Table 6 shows the comparison of soil carbon pool across different factors such as land use or forest types, altitude, and soil depth.

**Table 6.** The comparison of soil carbon pool across different factors.

| Location | Altitude (a.s.l) | Land Use/Forest Type | Soil Depth (cm) | Total Soil Carbon Pool (Mg C ha$^{-1}$) | Reference |
|---|---|---|---|---|---|
| Sabah, Malaysia | 1000–1600 m | Intact Forest | 30 | 96.42 | [2] |
|  |  | Logged-Over Forest | 30 | 91.14 |  |
|  |  | Plantation Forest | 30 | 88.92 |  |
| Singapore | 164 m | Primary Forest | 100 | 77.50 | [48] |
|  |  | Secondary Forest | 100 | 103.9 |  |
| Sabah, Malaysia | 300–470 m | Agroforestry system (Oil palm × Agarwood) | 30 | 39.12–49.75 | [49] |
|  |  | Monoculture plantation (Oil palm) | 30 | 43.09–45.46 |  |
|  |  | Natural Tropical Forest | 30 | 36.30 |  |
| Peninsular Malaysia, Malaysia | ~600 m | Unlogged Forest | 100 | 87.86 | [50] |
|  |  | Logged Forest | 100 | 65.55 |  |
|  |  | Rubber Smallholder | 100 | 67.50 |  |
|  |  | Rehabilitated Forest | 100 | 76.00 |  |
|  |  | Degraded Forest | 100 | 44.80 |  |
| Jiangxi Province, China | 127–1207 m | Bare Land | 100 | 19.86 | [51] |
|  |  | Restored Forest | 100 | 21.87–39.65 |  |
|  |  | Undisturbed Forest | 100 | 75.90 |  |
| Sabah, Malaysia | 300–600 m | Supervised logging with climber cutting | 100 | 87.95 | This study |
|  |  | Conventional logging | 100 | 71.17 |  |
|  |  | Virgin forest | 100 | 83.20 |  |

### 4.4. Total Soil, Organic Layer, and Litterfall Carbon Pool

The finding shows that soil contributed the highest total carbon pools among the three carbon pools (Table 5) studied in this research. Soil carbon pools can contribute up to 46% of the total carbon pool in a tropical forest ecosystem [4]. For example, Saner et al. [52], in

their study at Malua Forest Reserve, Sabah, Malaysia, reported that approximately 24% of the total ecosystem carbon pool in their study area was found in the soil carbon pools, while Besar et al. [49], in their study at Tawau Hill Park, Sabah, Malaysia, found that only 13% of the carbon was stored in the soil carbon pool.

The litterfall production in the forest is one of the most valuable indicators to show the forest's productivity and ecological functions, and to monitor the site productivity and its nutrient cycling [53]. It also acts as the primary route of carbon and mineral transfer from forest vegetation to the soil. In addition, it could provide shelter to organisms ranging from microbes to small mammals [54,55]. While all the other carbon pools do not show any significant difference in their mean across the different treatment plots and virgin forest plots, the opposite was observed in the mean of litterfall carbon. The result showed that the logging treatments have significantly decreased the litterfall carbon compared to the virgin forest areas. The values were $5.95 \pm 0.20$ Mg C ha$^{-1}$ in the supervised logging with climber cutting areas and $5.50 \pm 0.39$ Mg C ha$^{-1}$ in the conventional logging areas. The highest value was observed in the virgin forest areas, $7.64 \pm 0.6$ Mg C ha$^{-1}$.

The result published by Saner et al. [56] found no significant difference between the litterfall rate of the selectively logged and the unlogged forest. They also recorded a lower range of litterfall carbon compared to this study, which was between $4.8 \pm 0.1$ Mg C ha$^{-1}$ yr$^{-1}$ and 4.9 Mg C ha$^{-1}$ yr$^{-1}$. Another study that found a lower amount of litterfall carbon was Suhaili et al. [2], with a range of $3.05 \pm 0.13$ to $3.48 \pm 0.09$ Mg C ha$^{-1}$. The production of litterfall in the forest was controlled by a few factors, such as geographical location, climate conditions, and vegetation structure [57].

The litterfall production in the forest also positively correlates with soil carbon concentration. For example, Leff et al. [58] found that doubling the litterfall input could increase soil carbon concentration by up to 31% while removing it could reduce soil carbon concentration by 26% [59]. This fact shows the vulnerability of these pools to changes and how it also could affect another pool. Thus, it makes it essential for us to enhance our knowledge of litterfall production so better forest management plans can be implemented to maintain the forest condition and function as carbon pools.

### 4.5. Soil Respiration Rates in Logged Forest

Soil respiration is defined as the $CO_2$ efflux from the soil surface. It is also the sum of multiple processes that occur inside the soil, such as root respiration and decomposition of plant residues [7]. This finding shows that supervised logging with climber cutting has helped increase the soil respiration rate, while conventional logging has a more prominent tendency to decrease the rates. For comparison, Takada et al. [7] studied the changes in soil respiration rates after logging in an upper tropical hill forest in Perak, Malaysia. They found that the rates decreased by approximately 25% after the logging activity. This finding implies that implementing a suitable logging method and adding silvicultural treatment can help increase soil respiration rates rather than decrease them.

The percentage of soil organic matter greatly influences the soil respiration rate, thus explaining the higher rate of soil respiration in the supervised logging with climber cutting treatment [59]. The higher rate of soil respiration in those plots might be due to higher heterotrophic respiration, mainly because of the higher value of soil organic matter respiration [14]. A few publications were found that reported studies of soil respiration in tropical forests. For example, Adachi et al. [60], in their study at a primary and secondary forest in the Pasoh Forest Reserve, Negeri Sembilan, Malaysia, recorded higher soil respiration rates with mean values of 948 and 707 mg $CO_2$ m$^{-2}$ h$^{-1}$, respectively.

In a study of a lowland dipterocarp forest in East Sabah, Malaysia, Saner et al. [56] found that the soil respiration in the logged forest was higher compared to the unlogged forest. The rates were $28.6 \pm 1.2$ and $21.6$ mg C ha$^{-1}$ yr$^{-1}$. In a study of a lowland mixed–dipterocarp forest in Sarawak, Malaysia, Katayama et al. [61] found that the soil respiration rate was $246.47$ mg C m$^{-2}$ h$^{-1}$. They mentioned that the soil respiration rates at their study site correlated positively with increasing tree DBH. Furthermore, another study stated that

soil respiration has a positive correlation with the stand age, as it also means there is an increase in root biomass and accumulation of organic carbon [62].

In the study at Perak, Malaysia, Takada et al. [7] states that the soil respiration rates do not show any temporal changes after logging activity. The average soil respiration rate was 461.80 mg C m$^{-2}$ h$^{-1}$. Typically, soil temperature and water content were the most critical factors influencing soil respiration rates. However, according to some studies, both these factors have a lessened impact in tropical regions compared to the other regions, as the soil temperature in this region is relatively constant [7,60].

## 5. Conclusions

This research found that there was no significant difference observed in the soil properties across the different logging treatments and the virgin forest. The finding also shows that after 26 years of logging, the soil, organic layer, and litterfall carbon pools in the logged forest have recovered, as the amount of biomass production in all logged forests was not significantly different compared to the amount of biomass production in the virgin forest. The supervised logging with climber cutting logging treatment has a higher total mean soil carbon pool than virgin forest and conventional logging. Furthermore, the finding on soil respiration rates also shows a positive review of this logging method.

In short, implementing a suitable forest management method such as the supervised logging method, and combining it with silvicultural treatment such as climber cutting, seems to be able to help speed the recovery process of the carbon pools in the forest and increase soil respiration rates. However, as there no significant difference was observed in this finding, a more detailed study needs to be performed to answer the remaining research questions, such as how long does it take for the soil carbon to recover after logging and which logging method provides the fastest recovery rate. This can be done by increasing the replication of plots and conducting an early sampling, such as immediately after completion of the logging.

These findings highlight the importance of accurate and detailed quantification of the impact of different logging methods on the total ecosystem carbon pools inside the forest; thus, a better forest management plan can be implemented to maintain the forest condition and function as a carbon pool.

**Author Contributions:** Conceptualization, N.H., U.I. and E.P.A.; methodology, N.S.S., S.N.A. and D.L.; software, N.S.S. and S.N.A.; validation, N.A.B., N.S.S., N.H. and U.I.; formal analysis, N.A.B., N.S.S., S.N.A. and W.V.C.W.; investigation, N.A.B., N.H. and U.I.; resources, N.H., N.A.B. and U.I.; data curation, N.A.B., N.H. and U.I.; writing—original draft preparation, N.A.B., N.S.S., S.N.A. and W.V.C.W.; writing—review and editing, E.P.A., D.L. and U.I.; visualization, N.S.S. and S.N.A.; supervision, N.A.B., N.H. and U.I.; project administration, N.A.B., N.H. and U.I.; funding acquisition, N.A.B., N.H. and U.I. All authors have read and agreed to the published version of the manuscript.

**Funding:** This research was funded by the Swedish Research Council (FORMAS) through the Swedish University of Agricultural Sciences (SUAS) to Universiti Malaysia Sabah with grant numbers FORMAS-2016-20005 and GLA0020-2018. The APC was funded by Universiti Malaysia Sabah.

**Data Availability Statement:** Not applicable.

**Acknowledgments:** The authors want to sincerely thank the Yayasan Sabah Group and Sabah Biodiversity Center for permitting the research in the study area. The authors would also like to express gratitude toward the kind and hard-working people in INIKEA, Luasong, especially David Alloysius and all the research assistants, for their helping hand during the research. Lastly, the authors would like to specifically mention and thank Ignatius Baxter, Syahrir Mhd Hatta, Bryan, Ho Pui Kiat, Belleroy, Nicholson, Mohd. Ikram, Bonaventure, Buhaiqi, Rino Flemino, and Azzah for their extensive support during fieldwork data collection.

**Conflicts of Interest:** The authors declare no conflict of interest. The funders had no role in the design of the study; in the collection, analyses, or interpretation of data; in the writing of the manuscript; or in the decision to publish the results.

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
