# Peer review of "Soil Carbon Pool and Carbon Fluxes Estimation in 26 Years after Selective Logging Tropical Forest at Sabah, Malaysia"

_forests, doi:10.3390/f13111890_

Round 1

Reviewer 1 Report

Thank you for the opportunity to review the paper. The paper tried to address an important issue, but at this stage, it is not publishable and need major revision. My comments are suggestions are:

  • Title:
    • Change “selected” by “selective”
  • Abstract:
    • First sentence is controversial. Boreal forests store much higher amount of soil carbon than tropical rainforests.

  • Introduction:
    • Nicely written, but it is too long. The first three para are not relevant here and information given in these paragraphs are available in 100s of literatures. It will be better to talk more about the different facts of figures of tropical soils, selective logging, typical features of two types of loggings, and their impacts on soil carbon pool.
    • Links (and discuss) soil carbon with different sustainable development goals (SGDs).

  • Methodology
    • Study sites are nicely discussed, loved it
    • How can you make sure that the 8 sample plots represent the entire population? Throughout the paper, you need to scale down the tone. This research is good enough for setting hypothesis but not for policy making. Further research on broader area in needed to know whether this finding is broadly applicable.
    • Also, it is not clear how those 8 plots are selected, whether the process in statistically valid, how soil coring points are selected with the plot. Similar information for other carbon pools is needed.
    • What can you make sure that the soil carbon changes were solely due to logging? Taking a virgin unlogged forest as the control plot is not enough. Substantiate your claims with set of reasons.
    •  
  • Discussions
    • Discussion section is poor, some claims need to substantiate by literatures/citations
    • It is interesting to note that the logging with climber cutting (SLCC) has the highest soil carbon amount, even higher than control plots. This is hard to believe. A lengthy discussion of global literatures and suggestions of the list of reasons is highly recommended.  
    • What can you make sure that the soil carbon changes were solely due to logging? Taking a virgin unlogged forest as the control plot is not enough. Discuss and suggest possible limitations of the study.  
    • How your soil carbon compares with other similar forests around the world. Make a comparative table and discuss more.

  • Conclusions:
    • Considering above methodological issues, scale down your claims and highlight limitations of this study and future research areas.

Some of the relevant papers are given below. Discuss them in introduction and discussion section, where appropriate.

  • (2008). Comparing and predicting soil carbon quantities under different land use systems on the Red Ferrosol soils of Southeast Queensland, Journal of Soil and Water Conservation, 63 (4), 250-257.
  • (2014) Can vegetation types work as an indicator of soil organic carbon? An insight from native vegetation in Nepal. Ecological Indicators, 46, 31-322

Reviewer 2 Report

This study investigated physicochemical properties of forest soils 27 years after two different types of experimental logging (conventional logging and supervised logging with climber cutting). The study found no difference in the soil properties among the soils from the area with different logging treatments except litterfall carbon pool. Below are my comments and suggestions.  

            The main concern of this study is that the presentation of the statistical tests is either incomplete or not interpreted properly. In the text (mainly in the discussion), the authors mentioned difference in several soil properties with the SLCC and/or CL compared to the VF, but the only significant difference that I can find based on the provided tables was litterfall carbon pool. Furthermore, the letter ‘a’ for values statistically not different is on every value in one table and next to the first values on another table. If it’s not different, letter ‘a’ is not needed at all. Authors need to re-check the values in tables and correct the discussion accordingly.  

            In the method section, the information about the study area seemed not enough to understand the experimental design. Because the time difference between the logging treatment and the actual measurements is large, authors need to show that the established plots were comparable before the treatment or at the time of measurements. For example, species composition, tree density, leaf area index, or biomass in year 1 and year 27 can be used for an appropriate comparison among treatments.

            When authors mentioned findings of this study in the discussion, figure and table numbers need to be stated in the text to help readers to find the relevant information easily. Because this study includes various properties in the results and many of them was in tables, pointing out the location of the finding would be useful.

Minor comments:

L15: gasses à gases

L49: microbial à microbes

L50: 1 Gigaton = 10^9 ton

L56-73: use a consistent unit of mass (either Gt or Pg) in the introduction for better comparisons.

L85: remove ‘also’

L116: ‘area which, was’ à ‘area, which was’

L126: while à and

Figure 1: ‘C’ for control in panel c need to be replaced with ‘VF’ for consistency.

L154: need detailed definition of SLCC and CL along with key differences between the two logging methods. A table can be helpful.

L116-117: The meaning of the sentence is unclear.

L169: remove ‘also’

Table 1: What does the letter ‘a’ mean for the Bulk Density? It’s all ‘a’ and added only with the depth of 0-10. If the difference of Bulk Density in 0-10 cm depth among treatments is not significant, ‘a’ may not be needed. What about the values from other depths?

Table 2: same issue with Table 1.

L299: Why do you think it’s coincident. If the plots has similar species, their litterfall production could be similar.

L426: remove ‘production’

L428: remove ‘based on the result’

L504: not sure ‘in contrast with’ is a proper form in this sentence.

L532-537: Does this study shows microbial activity greater than current study?

Reviewer 3 Report

Two logging treatments are selected in the manuscript to study soil carbon pool and carbon fluxes estimation in 27 years tropical forest at Sabah, Malaysia. The logging treatments were supervised logging with climber cutting (SLCC), and conventional logging (CL). A virgin forest (VF) as an unlogged forest was used as the control plot. The topic is generally interesting, but the research significance is not outstanding in the manuscript. There are some suggestions in the writing.

1.      In the Abstract, please condense the research background and the introduction of materials and methods, and add the research results.

2.      The Introduction is large and general, and lacks very relevant research progress. The scientific purpose of the research is not clear, the author just mention that the results is helpful as the baseline data for the policymakers.

3.      How many soil samples of every layer are collected? How many repetitions?

4.      There are almost no significant differences of the statistical analyses in Tables 1 – 4. How does the manuscript get the conclusion that SLCC treatment decreases the soil bulk density and increases soil organic matter and carbon concentration?

5.      There are two parts of soil carbon pool and soil respiration rate in the manuscript. But it seems that their relationships are not analyzed. How do they interact?

6.      The discussion is too long and lacks perspective and comprehensiveness, which are all further analyses corresponding to various parts of the results. It’s long and unreadable.

7.      Line 356, with, not “whit”.

8.      Line 484, 4.4, not “3.4”.

9.      Line 518, 3.5, not “3.5”.

Round 2

Reviewer 1 Report

This paper has been improved a lot but the 8 sample plots cannot represent the entire population. Although they have scale down the tone, it is not enough. This research is good enough for setting hypothesis but not for policy making. Please revisit entire paper and change your tone. Second, please edit properly, there are some types and grammatical errors. 

    •  

Author Response

Dear Reviewer,

Thank you

Best regards

Normah

Reviewer 3 Report

The manuscript is totally improved after revision. Most of the tables are replaced with new tables that are lack of statistical analysis in table 1-4, how to get the conclusion that the difference is not significant in the Conclusion section? Please perform statistical tests on the data in the tables.

Some words in the manuscript are not smooth, please refine the full text.

Author Response

(The authors gave the same response as above.)
